# Semiautomated Statistical Discontinuity Analyses from Scanline Data of Fractured Rock Masses

**Christian Zangerl [1],\*, Matthias Koppensteiner [1] and Thomas Strauhal [2,3]**

[1] University of Natural Resources and Life Sciences, Vienna, Department of Civil Engineering and Natural Hazards, Institute of Applied Geology, Peter Jordan-Strasse 82, 1190 Vienna, Austria

[2] alpS GmbH, Grabenweg 68, 6020 Innsbruck, Austria

[3] ILF Consulting Engineers Austria GmbH, 6063 Rum, Austria

\* Correspondence: christian.j.zangerl@boku.ac.at; Tel.: +43-1-47654-87211

**Abstract:** Quantitative statistical discontinuity analysis of fractured rock masses is commonly applied in the fields of engineering geology, rock mechanics, slope stability, and hydrogeology. This study presents a workflow for the semiautomatic determination of basic discontinuity parameters, such as spacing, frequency, trace length, and termination, from scanline surveys written for the open-source software Octave. The aim is to provide theoretical background information and scripts for a quick introduction to all interested parties from academia and consulting, in order to promote the use of widely known and accepted statistical evaluation methods. Data from a study site in granodioritic rock are analyzed in the context of the proposed workflow. These test data and all scripts (m-files) used in the study are provided in order to minimize initial training time. The scripts provided herein are kept short and simple, but can be used as a basis for advanced automation of the workflow and presentation of the results.

**Keywords:** discontinuity analysis; scanline; spacing; frequency; trace length; GNU Octave; semiautomated work flow

## 1. Introduction

Quantitative discontinuity analysis of fractured rock masses (e.g., orientation, spacing, frequency, size and trace length, and surface conditions) is commonly needed for a wide variety of applications, comprising (i) rock mass characterization and classification for tunneling, mining, and slope design (e.g., [1–3]), (ii) discrete fracture network characterization and modeling for fluid flow in fractured rocks and geothermal energy extraction (e.g., [4–7]), and (iii) natural hazard assessment and slope stability (e.g., rockfall and rock avalanches, [8,9]). One parameter highly relevant for the fields mentioned above is in situ block size distribution (IBSD), which is also determined from discontinuity analysis methods [10].

Geometrical discontinuity data for rock mass characterization can be obtained by several methods, comprising scanline and window sampling [11–16], borehole imaging based on optical and acoustic televiewer methods [17,18], and logging of oriented borehole cores [19]. In the last few decades, remote surveying techniques that can acquire 3D digital data of a rock face with high resolution, in particular digital photogrammetry, light detection and ranging (LiDAR) and unmanned aerial vehicle (UAV), have become state of the art and are widely applied [20]. All of these methods have advantages and limitations that should be evaluated before application. Traditional field measurements based on scanline sampling require safe access to the rock face and rockfall-safe mapping conditions. In addition, the area of mapping is limited by the accessibility of the geologist, and by its nature the method is time-consuming. Advantageous is the close visual inspection of the rock face, which makes it possible to recognize (i) barely visible traces of discontinuities, (ii) discontinuity infillings or coatings, (iii) strength-relevant very thin intercalations of (e.g., clay) layers,

and (iv) the roughness and curvature. Moreover, the origin of a discontinuity, e.g., bedding plane, extensional joint with plumose structures, and fault plane with slickenside striations, can often be determined only by close visual observation.

Data analyses of scanline data are well established and underpinned by statistical methods [15,21]. Due its line-wise sampling characteristics, outcrop-based scanline data can be nicely complemented by borehole data if televiewer data or oriented borehole data are available. Given that scanline data are often subhorizontal and boreholes subvertically drilled, a good representation of the discontinuity network with minimized sampling bias can be obtained. However, the handmade and geological compass-based sampling at the rock face and subsequent data analyses are time consuming and thus considered expensive. In comparison to the costs of drilling, scanlines are quite affordable, especially if the evaluation process can be automated.

Areal-based remote sensing methods provide other advantages, such as (i) in situ measurements from a safe distance, (ii) 3D discontinuity measurements of large rock faces with high accuracy and high spatial resolution, (iii) measurement of a high number of discontinuities, (iv) determination of the areal extent of large discontinuities with a better representation of the orientation, (v) little time required for on-site measurements relative to the rock face area being sampled, and (vi) fully automatic analyses with little effort of the processor. However, these methods also have disadvantages, comprising (i) small discontinuity surfaces and closed traces are often neglected, (ii) characteristic discontinuity surface features, such as coating, slickenside striation, plumose structure, or roughness cannot be identified, (iii) the type of the discontinuity cannot be determined, and (iv) difficulties in determining representative orientation of wavy or curved discontinuities.

Even though the scanline method has been known since the seventies and is well suited for the combination with other linear explorations (drilling), it still has not become a standard and is rarely performed in commercial engineering geological or geotechnical studies. The reason for this lies in the assumption that scanline mapping and analysis is time-consuming and consequently costly compared to the conventional but much more subjective outcrop surveys. The conventional outcrop survey method is understood as the recording of discontinuity parameters on processor selected individual outcrops, usually by assigning them to classes (e.g., spacing classes). This way of presentation can lead to difficulties in the subsequent application for quantitative geotechnical interpretations, and can be subjective and not representative in some cases.

The aim of this study was the development of a workflow and tools written for the software GNU Octave ([22], which is a free software licensed under the GNU General Public License (GPL). Although not tested, MATLAB [23], which is compatible with Octave, can probably be used to compute the m-scripts presented here. Our workflow provides a semiautomated statistical discontinuity analysis of raw scanline data for the determination of (i) the normal set spacing probability density distribution and the mean of individual discontinuity sets, (ii) the mean linear frequency of sets, (iii) the mean and the distribution of trace lengths of each sets, and (iv) the termination index [14]. The tools developed in this study are based on Octave m-scripts, which after executing provide an output for the implementation in numerical modeling software applications (e.g., distributions of joint orientation and normal spacing for various discrete fracture network (DFN) modeling, for blasting and rockfall evaluations (e.g., IBSD), or for rock mechanical calculations (e.g., GSI). In order to make it as easy as possible to get started using these m-scripts, only the main calculation steps have been implemented and complete automation has been omitted. A major goal of our study was to provide a low-threshold entry point for anyone interested in performing discontinuity analyses without requiring a large investment of time and to further support the wider application of the method in engineering geology or geotechnical engineering. If required, https://www.octave.org/support (accessed on 1 March 2022) will help with the programming language of Octave for adaptation of scripts to certain requirements. The workflow is presented and tested exemplarily by using data

from a case study with extensive scanline surveys in the Austrian Alps, composed of granodioritic gneisses.

## 2. Methods and Theoretical Background

### 2.1. Data Acquisition Procedure

The scanline discontinuity sampling method is based on installing a measuring tape (usually 10 to 30 m in length) on a clean and (as much as possible) planar rock face that is large relative to the size and spacing of the discontinuities exposed [14]. Such rock slopes are found on natural outcrops, road cuttings, quarries, open pit mines, and unsupported adits. Locating planar outcrops in the field large enough for scanline mapping can sometimes be difficult and limit the applicability of this method. Priest [14] recommends approx. 150–350 measurements to characterize a homogeneous area of a rock mass. Zeeb et al. [11] defines 225 as the minimum number of required discontinuity measurements. In order to avoid large errors in calculating the spacing values, the orientation of one single scanline should not deviate more than 20° from a straight line [14]. If this is not possible due to the irregular outcrop situation, the scanline should be divided into sections. For a representative data acquisition, it is important to align multiple scanlines at different orientations to provide a three-dimensional sampling setup of the discontinuity network and to minimize the orientation sampling bias. Since scanlines are predominantly subhorizontally oriented, combined analyses with measurements from steeply inclined boreholes, if available, can be performed. Orientation and spacing data of boreholes can be gained from acoustic and optical televiewer borehole imaging. By its nature, it has to be mentioned that borehole data do not provide any information concerning the size or trace length of discontinuities. Core logging data are only useful if special sampling and analysis techniques are applied to enable the determination of the true orientation of the discontinuities.

Concerning the scanline sampling procedure, the scanline tape is visually inspected for each intersection of a discontinuity trace with the tape [14]. In order to avoid unnecessary data processing, it is advantageous to place the tape on the rock face in such a way that the first discontinuity is located at the zero marker point of the tape. Concerning the sampling procedure, each discontinuity trace and its properties intersecting the measuring tape are recorded systematically on a logging sheet (Figure 1). These properties include the intersection distance, dip direction and angle, semi-trace length above and below the scanline, termination above and below, as well as comments. Optionally, additional parameters such as roughness, curvature, mineralization, opening width or strata thickness can be recorded according to proposed empirical classification schemes (e.g., [1,14]). Comments include the type of discontinuity, infillings and coatings based on mineralization. Mandatory for the analyses of the data are measurements of the trend and plunge of the scanline.

| Scanline Information | Outcrop information | |
|---|---|---|
| No. 1 | Location: Austria | Height: 3 m |
| Name: CZ | Dip direction: 346 | Width: 20 m |
| Date: 1.5.2022 | Dip angle: 85 | Excavation method: natural |
| Trend of Scanline: 170 | Overhanging slope (yes/no): no | Condition of exposure: weathered |
| Plunge of Scanline: 10 | Rock type: Granodiorit gneiss | Comments: no |

| Intersection distance [m] | Dip direction [°] | Dip angle [°] | Semi-trace length above [m] | Semi-trace length below [m] | Termination above | Termination below | Comments |
|---|---|---|---|---|---|---|---|
| 0 | 270 | 86 | 2.5 | 1.4 | 2 | 1 | Joint |
| 0.1 | 272 | 75 | 3.4 | 1.5 | 2 | 1 | Joint |
| 0.6 | 10 | 88 | 1.5 | 1.0 | 3 | 2 | Joint |
| 0.9 | 135 | 8 | 0.9 | 0.8 | 3 | 2 | Joint |
| 1.1 | 262 | 75 | 1.2 | 1.3 | 2 | 2 | Joint |
| 1.3 | 82 | 12 | 1.0 | 1.1 | 2 | 3 | Joint |

**Figure 1.** Scanline survey logging sheet (modified after [14]).

### 2.2. Theoretical Background

#### 2.2.1. Determination of Discontinuity Sets

The first step of the statistical discontinuity analyses focuses on the determination of the discontinuity sets of the rock mass. Given that several commercial or freeware-based software products with implemented clustering algorithms are available, a new script was not developed in this study. Almost all of these products allow the import and export of tabulated data, such as the scanline logging sheet presented herein. Some of these products are able to define discontinuity sets by advanced statistical methods. Markovaara-Koivisto and Laine [24] proposed a discontinuity set clustering algorithm based on the K-means algorithm for clustering of the normal vectors of all joints, with the possibility of manual adaptation of individual discontinuity assignment subsequently. This method requires the information of the number of discontinuity sets as a prior input. Pecher [25] developed an expert-supervised method for grouping of orientation data that requires the visual assignment of initial values for the mean orientation of one to a maximum of seven clusters. The fully automatic clustering method of Klose et al. [26] is based on vector quantization. Similar to other methods, the number of clusters has to be defined in advance.

However, all of these methods assign discontinuities to discontinuity sets and calculate their mean solely on the statistical distribution of the measured orientation data. Information regarding the genesis of the discontinuity set is not considered, but can be helpful for the discrimination of discontinuity sets [27]. Overlapping clusters of discontinuity sets are common and often related to multiphase fracturing processes, indicating that within a single discontinuity set not all the discontinuities are of the same age. For example, a comparison of the linear discontinuity frequency of the main joint set in a granitic rock mass measured at ground surface and in an investigation drift 550 to 1250 m below the surface show a decrease in frequency by a factor of 3.6 [28]. The depth dependence of joint density indicates that additional discontinuity formation processes near the ground surface took place, most likely due to stress release by gravitational unloading. According to Hancock et al. [29], the complex regional and local stress field and its alteration over time may lead to the formation of a joint continua that is characterized by a coaxial angular continuum of fracture planes, enclosing a maximum angle of 45° and including extension and hybrid shear fractures (i.e., failure in the shear extension–fracture transition). Under such conditions, defining a well-developed and delineable discontinuity set is difficult, as a large variation of orientation measurements and overlaps with other sets occur. This results in discontinuity orientation clusters with a highly dispersed scatter around a mean. From the rock mechanical aspect, it is useful to discriminate the discontinuity sets not only on the basis of the statistical distribution of the orientation but also on structural-geological criteria related to the formation processes (i.e., sedimentary bedding, foliation planes, stress-induced joint and shear fractures), which may help delimiting sets. Numerous studies show that a maximum of 3–5 discontinuity sets are sufficient in many cases to characterize a rock mass (e.g., [2,28,30]). A high number of sets may be obtained by the unfavorable delimitation of structural homogeneous areas and should be avoided, because it can lead to a too low number of discontinuities for each set for the subsequent analyses.

#### 2.2.2. Determination of the Total Spacing and Normal Set Spacing

Discontinuity spacing is a basic parameter that affects the size of the blocks and thus rock mass strength, deformability, and hydraulic conductivity. Priest [14] differentiated three types of discontinuity spacing: (i) total spacing, (ii) set spacing, and (iii) normal set spacing. Total spacing is the spacing determined between a pair of immediately adjacent discontinuities, measured along a defined scanline. Set spacing is the spacing between a pair of immediately adjacent discontinuities from a particular discontinuity set, measured along a defined scanline. Normal set spacing is the spacing that is determined along a scanline oriented parallel to the mean normal to the set. Whereas the former two types of spacing parameters are scanline direction-dependent, normal set spacing is a general rock mass parameter. The normal set spacing $X_n$ of a set is calculated on basis of the

measured distance between two discontinuities of the same set along the scanline $X_d$ and the angle $\delta$ between the orientation of the normal vector of the discontinuity and the scanline orientation [14,31]:

$$X_n = X_d \cos \delta \tag{1}$$

where $\delta$ is the acute angle between the discontinuity set normal and the sampling line and may be expressed as:

$$\delta = |\cos(\alpha_n - \alpha_s) \cos \beta_n \cos \beta_s + \sin \beta_n \sin \beta_s| \tag{2}$$

where $\alpha_n / \beta_n$ are the trend/plunge of the normal set vector and $\alpha_s / \beta_s$ are the trend/plunge of the scanline. A limitation of the maximum angle between scanline and the orientation of the normal vector of the set can be set to avoid extreme normal set spacing values [32]. The mean normal set spacing $\overline{X}$ is determined by:

$$\overline{X} = \sum_{i=1}^{n} X_i / n \tag{3}$$

where $X_i$ is the $i$-th spacing value and $n$ the total number of spacing values.

The discontinuity spacing values from scanlines can be visualized by histograms, which indicate if the intersection points of traces along the scanline are evenly spaced, clustered, or randomly distributed. Hudson and Priest [21] proposed studying the pattern of spacing values by applying density probability functions. Based on several case studies, they observed that randomly located intersection points of traces along a scanline produce a negative exponential probability density distribution of spacing values:

$$f(x) = \lambda e^{-\lambda x} \tag{4}$$

where $f(x)$ is the probability density $\lambda$ is the mean linear discontinuity frequency, and $x$ is the spacing value. Furthermore, they showed that the superimposition of different density distributions, such as negative, uniform, or normal distribution, will also lead to a negative exponential density distribution. Hence, for many situations, especially when the formation of a set is related to multiphase fracturing processes, it is likely that the measured distribution will coincide with a negative exponential form. However, in some cases, other density probability functions, for example, the log-normal distribution or the Weibull distribution may better represent the measured data set [28,33].

### 2.2.3. Determination of the Linear Frequency

One advantage of the negative exponential density distribution is that there is a direct relationship between the mean normal set spacing $\overline{X}$ and the linear frequency $\lambda$ of a set.

$$\lambda = 1/\overline{X} \tag{5}$$

If the linear discontinuity frequency of a set $\lambda$ is available, the frequency in any arbitrary direction can be calculated by $\lambda_s = \lambda |\cos \delta|$. For the case that several sets are present labeled by the notation $i$ the total frequency in any direction can be estimated by

$$\lambda_s = \sum_{i=1}^{n} \lambda_i |\cos \delta_i| \tag{6}$$

The linear frequency $\lambda$ is commonly referred to as linear fracture intensity $P_{10}$ [11]. Assuming a scanline orientation parallel to the normal of a discontinuity set, i.e., oriented equal to the mean normal set spacing, the relationship between linear ($P_{10}$), areal ($P_{21}$), and volumetric ($P_{32}$) fracture intensities is given by

$$P_{10} = P_{21} = P_{32} \tag{7}$$

### 2.2.4. Determination of Trace Lengths and Termination

A systematic mapping of entire discontinuity surfaces is difficult, since fully exposed surfaces are rare and termination characteristics are mostly not visible in 3D. Advanced remote sensing methods based on UAV and terrestrial laser scanning (TLS) provide some improvement of the sampling situation, since areal sampling of large rock faces is performed and discontinuities that are aligned subparallel to the rock face are scanned preferably. However, discontinuities exposing as traces are underestimated by these methods. Scanline mapping can reduce this problem, but has to account for other sampling biases, resulting from (i) the tendency to intersect preferentially the longer traces, (ii) trimming of short traces, i.e., the difficulty in measuring discontinuity traces with a length below a certain threshold value, and (iii) curtailment of long traces [14,15]. Some authors assume that discontinuities from a given set can be represented by circular disks where their diameters are distributed according to some probability density distribution. In order to avoid developing a model for the discontinuity shape, some authors have decided to treat the problem entirely in 2D [14,15,34]. Such 2D approaches are based on probability density distributions of traces, e.g., negative exponential, uniform, or triangular [14,15], whereas others are distribution-independent [16]. Herein, four different approaches to estimate the mean full trace length $\mu_L$ of a rock face for a set are presented, according to [14].

Approach I is related to considerations of [14,15] and is based on the assumption of a negative exponential distribution of full trace lengths for a given discontinuity set over an entire rock face with the mean $\mu_L$. Hence, the probability density distribution $g(l)$ with its mean $\mu_{gL}$ of the full trace lengths intersected by the scanline is given by:

$$g(l) = \frac{l\, e^{-(l/\mu_L)}}{(\mu_L)^2} \tag{8}$$

$$\mu_{gL} = 2\, \mu_L \tag{9}$$

Equation (9) indicates that the mean full trace length of the entire face is equal to half of the mean full trace length of the scanline intersected traces.

Approach II determines the mean of the full trace length distribution by considering the numerical proportion of discontinuities that have a semi-trace length less than the user-specified curtailment value $c$ [14,15]. Given that the total sample size of semi-trace lengths for a given discontinuity set is $n$ and a certain number $r$ of these have a semi-trace length less than $c$, then the mean of the full trace length $\mu_L$ can be calculated by:

$$\mu_L = \frac{c}{-ln\left(1 - \frac{r}{n}\right)} \tag{10}$$

Approach III is independent of the type of probability distribution and is based on a histogram of semi-trace lengths, i.e., number of values $N(l)$) and its intercept with the vertical axis at the limit $l = 0$ of the horizontal axis [14]. Therefore, a curve has to be fitted to the histogram and extrapolated to $l = 0$. Taking this interception value $N(l)$ the arbitrary defined class interval $\Delta$ and the total number of data $n$ the mean $\mu_L$ of the full trace length is expressed by:

$$\mu_L = \frac{n\, \Delta}{N(l)} \tag{11}$$

It has to be mentioned that the result is sensitive to the quality of the histogram and the assumed shape of the best-fit curve. In order to avoid too subjective an estimation of the mean trace length, we implemented a 6th-order polynomial function to fit a curve to the histogram.

Approach IV is based on Laslett [35] and accounts for various linear sampling biases, such as curtailment and edge effects. His approach does not require any information about

the probability density distribution and considers the measured full trace lengths as well as the termination characteristics. The mean full trace length $\mu_L$ is calculated by:

$$\mu_L = \frac{\sum_{i=1}^{n} x_i + \sum_{j=1}^{m} y_j + \sum_{k=1}^{p} z_k}{2n + m} \tag{12}$$

where $x_i$ is the $i$-th full trace length of $n$ traces with both ends visible, $y_j$ is the $j$-th full trace length of $m$ traces with one end visible, and $z_k$ is the $k$-th full trace length of $p$ traces with none of the ends visible.

### 3. The Discontinuity Analysis Workflow

The work flow of the discontinuity analyses is grouped into six steps (Figure 2). With the exception of step one, all calculations are performed using the open-source software Octave [22].

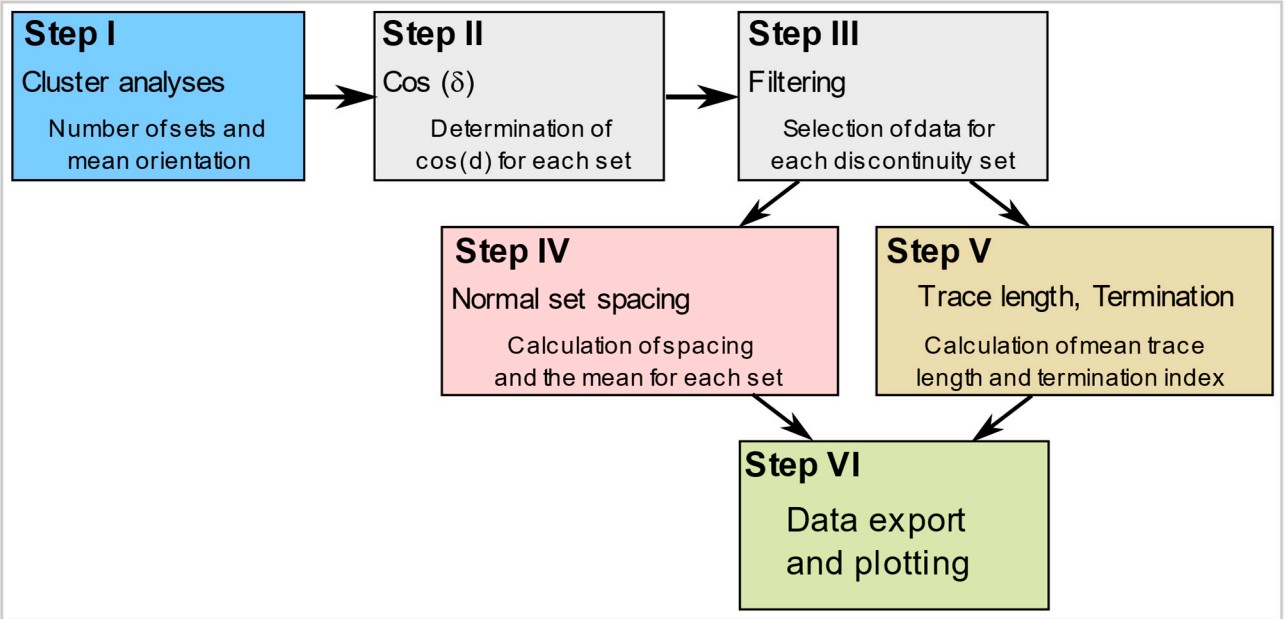

**Figure 2.** Workflow of the discontinuity analysis procedure, grouped into six steps.

Step I is related to the cluster analyses of the data record in order to identify and delimitate individual discontinuity sets. Due to the large number of existing free or commercial software products, e.g., Dips [36] and Stereonet [37], and their implemented individual methods for cluster analyses, it is recommended to perform Step I with those products. For the first step, all sampled data of a selected study site (assuming a homogenous rock mass domain) with multiple scanline sections are used. Whereas the most basic cluster analyses are based on a lower hemisphere contouring plot visually showing the individual sets, more advanced products use their implemented statistical clustering algorithms. As a result, the mean orientation for each set and a table of discontinuity numbers assigned to each set will be obtained. For the subsequent calculation, a consecutive number is assigned to each discontinuity of the raw data set.

Step II is necessary to determine the acute angle between the scanline orientation and mean orientation of each set for all selected scanlines. As input, a six-column table comprising set number, scanline number, scanline trend, scanline plunge, mean set dip direction, and mean set dip angle is loaded. After executing the Octave script (m-file *calculation_cos_delta.m*), the table is completed by three columns, i.e., trend and plunge of the normal to the mean set and the value of cos ($\delta$).

Step III is performed to filter and separate the spacing data for each set from the full table of spacing values according to a table of discontinuity numbers for each set

(m-file *spacing_filtering.m*). As a result, a 3-column text file comprising the discontinuity number, the scanline number and the intersection distance along the scanline is generated and saved.

Step IV is executed to calculate the normal set spacing values and the mean normal set spacing, as well as the mean linear frequency for all selected scanlines (m-file *normal_set_spacing.m*). As input, the following data are required: (i) the total number of the scanlines, (ii) the assigned number of the scanlines, (iii) the data file for the selected discontinuity set, and (iv) the mean cos ($\delta$) values for each scanline. After executing the script the intersection distances are multiplied by the mean cos ($\delta$) value for each scanline and then the difference between each value in the column is calculated, i.e., resulting in one-column matrix of set spacing and normal set spacing values. In addition, for each scanline, the mean set spacing and mean normal set spacing as well as the mean frequency for both is determined.

In Step VI, the resulting column vector of normal set spacing values is visualized by histograms and a negative exponential probability distribution is fitted to the data (m-file *spacing_histogram.m*).

Step V is performed to calculate the full trace length over the entire face and the mean full trace length according to the four selected methods mentioned above. The Octave script *trace_length_calculation.m* loads the trace length data file, which is structured as a six-column matrix of discontinuity number, scanline number, trace length above, trace length below, termination above, and termination below. By executing the m-file, the full trace lengths of measured data are determined by summing columns 3 and 4 (i.e., the semi-trace length above and below), and written in a new column 7. In addition, the mean full trace lengths and the termination indices are calculated.

In Step VI, the data record of semi- or full trace lengths intersecting the scanline is visualized by a user-specified histogram (m-file *trace_length_histogram.m*).

## 4. Application Example

### 4.1. Geological and Geographical Situation

Scanline surveys in granodioritic gneisses were made to collect discontinuities for rock mass characterization in a tributary valley of the Ötztal valley (Austria). Geologically, the fractured granodioritic gneisses belong to the Ötztal-Stubai basement complex [38]. Eight scanlines with a total of 596 discontinuity measurements were sampled at outcrops at altitudes of about 2150–2450 m.a.s.l. (Figure 3). Given that only scanlines of a selected geological domain were taken, i.e., only those located in the granodiorite gneiss unit, the numbering of the scanline does not correspond to a consecutive sequence. Scanlines labeled as 2, 5, 7, 8, 11, 12, and 14 are outside the area of interest and therefore neglected. However, a new labeling of the scanlines is not necessary, because the proposed Octave m-files do not require consecutive numbering. The scanlines are oriented in various directions and are located along planar surfaces of the rock walls. Due to inaccessibility and the difficult outcrop situation of the study area, it was a challenge to find favorable and large-enough scanline outcrops. However, locations were selected to enable sufficient scanline lengths ranging from 9 to 28 m. As lower limit, all discontinuities with trace lengths of at least 10 cm intersecting the scanlines were recorded. Parameters according to Section 2.1 were measured and recorded. Termination characteristics of the upper and lower ends of the discontinuity are recorded as (i) termination not visible (labeled as 1), (ii) termination in intact rock (labeled as 2), and termination at another discontinuity (labeled as 3).

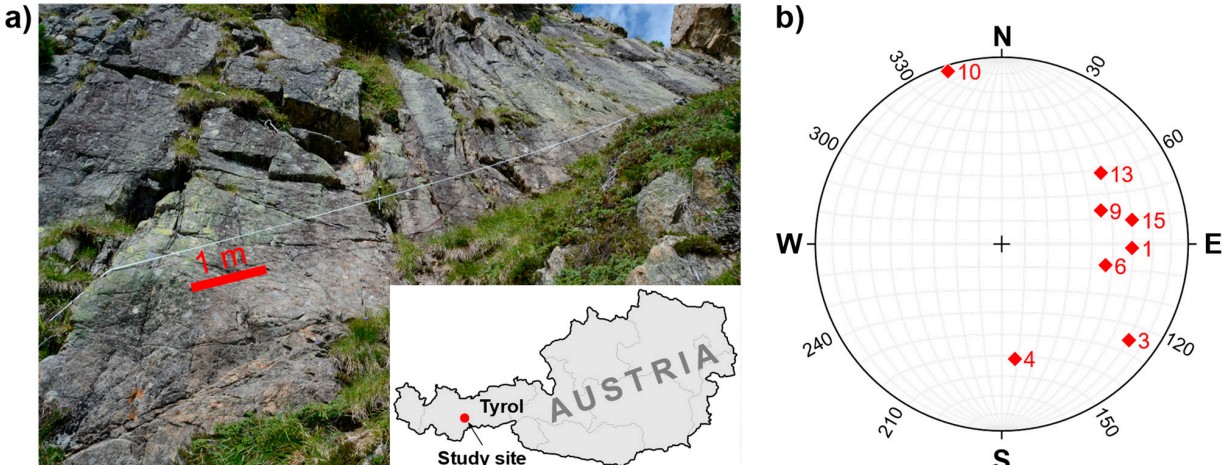

**Figure 3.** (**a**) Granodioritic rock face with fixed scanline tape, and (**b**) trend and plunge of the eight scanlines labeled as 1, 3, 4, 6, 9, 10, 13 and 15. Pole plots were created using the software Stereonet [37,39].

### 4.2. Discontinuity Orientation and Number of Sets

Concerning the fractured granodioritic rock mass, three main discontinuity sets were defined based on a contouring plot made by the software Stereonet [37] and additional spatially distributed outcrop measurements that supported the identification of the main sets. This means that the assignment of discontinuity sets was not only based on pure statistical criteria but also on geological observations in the field. The three sets show a mean dip direction and dip angle of 264/83 for set 1, 014/85 for set 2, and 254/18 for set 3 (Figure 4). Set 2 represents discontinuities (i.e., joints) that are aligned subparallel to the main foliation of the granodioritic rock mass. Output of this cluster analysis is a column vector showing the advancing number of discontinuities belonging to a selected set. The acute angle between the mean orientation of the discontinuity sets is calculated by equation (2), yielding 70.9° between sets 1 and 2, 65.3° between sets 1 and 3, and 85.9° between sets 2 and 3, indicating an orthogonal discontinuity network to a certain extent.

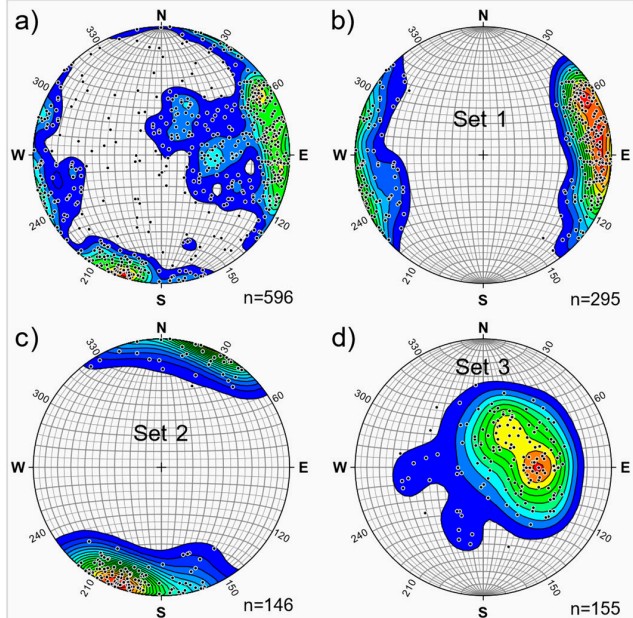

**Figure 4.** Discontinuity orientation data and clustering into three sets: (**a**) all measured data, (**b**) set 1, (**c**) set 2, and (**d**) set 3. Pole and contour plots were created using the software Stereonet [37,39].

### 4.3. Spacing and Frequency

According to the three sets defined by the cluster analysis, the normal set distribution and the mean of each set were determined by executing Steps II, III, and IV (Figure 2).

During Step II, calculation of the acute angle between the mean set orientation of each discontinuity set and each scanline orientation is performed, resulting in a table of cos (δ) values. For our case study, three discontinuity sets (i.e., set 1, 2 and 3) were defined from the data set based on eight scanlines, labeled 1, 3, 4, 6, 9, 10, 13, and 15 (see scanline_set_orientation.dat). By executing the m-file *cos_delta_calculation.m*, cos (δ) values are calculated for each set and stored as workspace variables *data_cos_set_1*, *data_cos_set_2* and *data_cos_set_3* in the mat-file *data_cos_set1-3.mat*.

In Step III, scanline intersection distances measured for a certain discontinuity set are filtered from the total data set of the entity of all scanlines (i.e., dat-file *spacing_data.dat*). Concerning the three sets of the test site, the filter criteria are loaded from the files, referred to as *set1_data.dat*, *set2_data.dat*, and *set3_data.dat*. Executing the m-file *spacing_filtering.m* saves the results in the mat-file *spacing_set1-3.mat*, including the variables for each set, labeled *spacing_set1*, *spacing_set2* and *spacing_set3*.

Step IV is carried out for each discontinuity set, i.e., sets 1, 2 and 3, separately and calculates the normal set spacing values. Before executing the m-file *spacing_calculation_set.m* for each set some modifications of the m-script have to be done. This means that the discontinuity set for which the calculation should be performed has to be selected by choosing the right variable, e.g., *spacing_set1*. Furthermore, the cos (δ) values obtained from Step II for the respective set must be entered in the m-file. In the case of the test site herein, for each discontinuity set, eight cos (δ) values should be implemented as a line vector in the same order as the scanline numbers above.

By executing the m-file *spacing_calculation_set.m*, the file *spacing_set1-3.mat*, which was determined in Step III is loaded and the variable *spacing_set1* for set 1 is selected. As a result, the spacing values are calculated for each set. The script determines several parameters that are stored in the following variables, comprising (i) a table of set spacing (*set_spacing_table*) and normal set spacing (*normal_set_spacing_table*) values, (ii) a table of mean set spacing and mean normal set spacing values for each scanline, and (iii) a table of the mean set and normal set frequency for each scanline. Finally, all results are stored in the mat-file *set1_spacing_parameters.mat*. This procedure must be repeated for sets 2 and 3.

In Step VI, the results of the spacing analyses are visualized as a histogram showing the normal set spacing versus frequency. The m-file *spacing_histogram.m* includes all calculations proposed by Priest [14] to plot a histogram with a negative exponential probability density distribution related to the unit width of the class interval. Figure 5 shows the results of the normal set spacing analyses for sets 1, 2, and 3 based on the data of the case study.

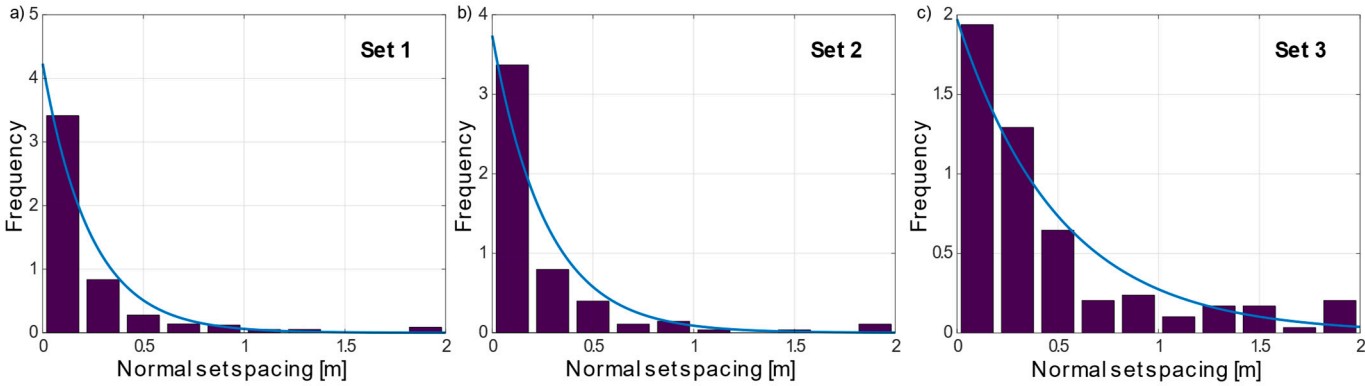

**Figure 5.** Frequency histogram versus normal set spacing with fitted negative exponential probability density distribution (blue line): (**a**) mean spacing and standard deviation of set 1 $\overline{X} = 0.236$ m, $SD = 0.369$, (**b**) set 2 $\overline{X} = 0.268$ m, $SD = 0.448$, and (**c**) set 3 $\overline{X} = 0.507$ m, $SD = 0.768$. Each of these three plots was created by the m-file *spacing_histogram.m*.

### 4.4. Trace Length and Termination

In Step V, the mean trace length of discontinuities for all data or each discontinuity set can be determined by four different approaches as per Section 2.2.4. These approaches are implemented in the m-file *trace_length_calculation.m*, providing an outcome of four mean trace length values. Before executing the m-file *trace_length_calculation.m*, the source data file has to be specified (e.g., for set 1, *trace_data_set1.dat*), containing six columns comprising the discontinuity number, scanline number, trace length above, trace length below, termination above, and termination below. In addition, a curtailment value *c* should be defined as proposed by approach II, and a histogram range and class interval should be defined for approach III, respectively.

After executing the m-file, the four different mean trace length values based on approaches I to IV are determined and saved in a mat-file (e.g., for set 1, *trace_termination_set1.mat*). In addition, a termination index is calculated from semi-traces above the scanline (variable *termination_above*), from semi-traces below the scanline (*termination_below*), and from all measured semi-traces (*termination_all*) for each set. The representative termination index value is usually gained from all measured semi-traces; however, the values above and below are helpful to validate the impact of concealed discontinuities. Given that in most cases, subhorizontal scanlines are measured, it can be assumed that below the scanline, a larger number of discontinuities are concealed, resulting in an underestimation of the true termination index.

Table 1 shows the results for the mean full trace length and termination analyses obtained by the m-file *trace_length_calculation.m.* The analyses indicate a remarkably large variation of the mean full trace length, depending on the applied approach. Approach I gave the smallest and approach III the highest values. Results of approaches II and IV are in between, with approach IV being smaller by 15% to 24%.

**Table 1.** Results of trace length and termination index analyses for sets 1, 2, and 3.

| Applied Approach for Mean Trace Length [m] and Termination Index Calculation [%]. | Set 1 | Set 2 | Set 3 |
|---|---|---|---|
| Approach I | 0.515 | 0.710 | 0.587 |
| Approach II | 0.770 | 1.160 | 0.969 |
| Approach III | 1.040 | 1.580 | 1.163 |
| Approach IV | 0.657 | 0.955 | 0.737 |
| Termination—above the scanline | 31.5 | 18.5 | 21.9 |
| Termination—below the scanline | 27.8 | 12.3 | 21.9 |
| Termination—all semi-traces | 29.7 | 15.4 | 21.9 |

In Step VI, the results of the trace length analyses are visualized as a histogram showing the semi-traces versus frequency. Functionally, the m-file *trace_length_histogram.m* is similar to the m-file for plotting the spacing histogram. Figure 6 shows the histograms of all semi-traces (i.e., above and below of the scanlines) for sets 1, 2 and 3 based on the data of the case study. It is worth mentioning that the negative exponential function based on the mean trace length of Approach I maps the distribution of lengths quite well.

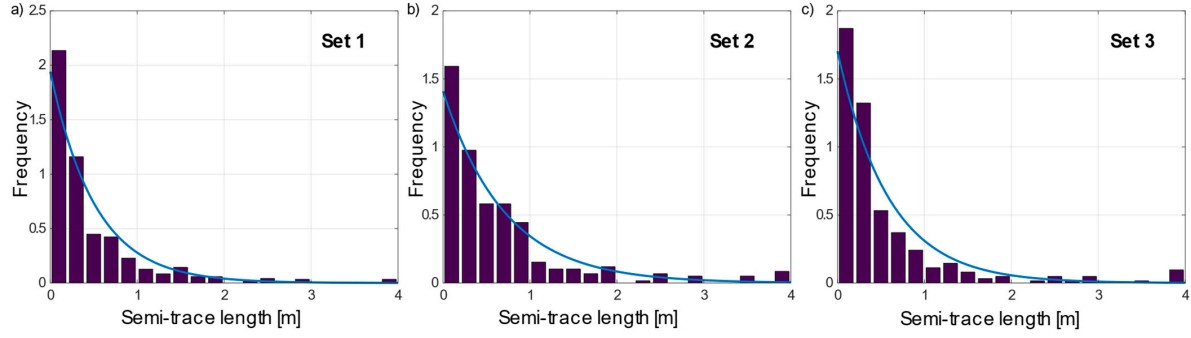

**Figure 6.** Frequency histogram versus semi-trace lengths with fitted negative exponential probability density distribution (blue line): (**a**) set 1 $\mu_L$ = 0.515 m, (**b**) set 2 $\mu_L$ = 0.710 m, and (**c**) set 3 $\mu_L$ = 0.587 m.

## 5. Discussion

The basic statistical discontinuity parameters that can be determined by the workflow and the scripts presented herein, comprising (i) normal set spacing for each set, (ii) linear frequency for each set, (iii) semi-trace length distribution and mean trace length of each set based on several approaches, and (iv) termination index of each set, are used for a wide variety of practical applications. These parameters serve as input for discontinuity characterization in the field of rock mechanics, groundwater flow, reservoir engineering, and geothermal energy extraction in fractured rocks.

For applications in rock mechanics, these parameters are essential to characterize the degree of fracturing and fragmentation of a rock mass. The volumetric joint count ($J_v$, [40]), a key value to describe the degree of jointing of a rock mass, represents the number of discontinuities intersecting a volume of 1 m$^3$ and is calculated by summation of the inverse of the mean normal set spacing of each set. The mean block volume, ($V_b$), of a rock mass that is fractured by three sets is determined from the discontinuity set orientation and mean normal set spacing [1,40]. The in situ block size distribution (ISBD) can be determined by empirical or numerical approaches based on statistical discontinuity data (e.g., [10,41,42]. Finally, the theoretical rock quality designation (RQD) value of the rock mass in any arbitrary direction is calculated from the linear frequency of the sets [14].

For the assessment and characterization of the rock mass properties in underground construction and slope stability, empirical methods based on rock mass failure criterions [43,44] and rock mass classification systems were developed [45]. All of these methods require data on the discontinuity network.

Due to the increasing computing power of PCs and workstations, the discrete fracture network (DFN) approach offers new possibilities to simulate the geomechanical and hydrogeological behavior of a rock mass with reasonable effort [5]. Applications related to slope stability, open pit mining, tunneling, dam foundations, groundwater flow in fractured rocks, radioactive waste disposals, and geothermal energy extraction are common and state of the art. Beside others, stochastically generated fracture networks are by far the most widely used DFNs for numerical modeling studies. Statistical discontinuity data from surface and subsurface scanline sampling and borehole imaging serve as the input for these models.

However, it should be noted that a statistical analysis of discontinuity data is also based on assumptions and limitations [14]. A key assumption is that the discontinuities are plane surfaces in 3D models and straight lines (traces) in 2D models. Accurate measurements of trace lengths are challenging due to the outcrop situation and accessibility. The real shape of discontinuity surfaces can hardly be determined at the outcrop and certainly not statistically recorded. Planar disks or polygons are therefore used as idealized shapes for the analyses and modeling. According to Priest [14], the normal set spacing is measured along a line oriented parallel to the mean normal to the set. Therefore, in the workflow presented here, it is assumed that the discontinuities of a set are aligned in parallel for the spacing calculation (i.e., cos ($\delta$) remains constant for a set and a selected scanline), leading to over- and underestimations of the distance. However, in comparison to RQD or total spacing, the normal set spacing value of each set of a rock mass is direction-independent. Statistical discontinuity analyses and their implementation into DFN models consider the geometrical parameters as independent random variables belonging to certain probability density distributions, such as negative exponential, normal, lognormal, gamma, or power law distributions [5]. The highly complex temporal and spatial geological evolution of a study area can only, if at all, be rudimentarily captured by such statistical discontinuity analyses. Methodically, great difficulty arises in generating a 3D fracture network model based on 1D or 2D measurement data. Scanline data or borehole imaging data are generally biased by censoring and truncation effects. Concerning accuracy and precision of discontinuity spacing estimates, e.g., due to short scanlines or small samples, the reader is referred to [14,33], and for discontinuity size estimations to [14,35].

　　　Despite the summarized assumptions and limitations, the workflow presented herein offers the opportunity of a more objective data acquisition and analysis compared to the classical outcrop sampling method with a subjective selection of recorded discontinuities and estimations of spacing and trace length parameters.

## 6. Conclusions

　　　A workflow for the semiautomatic determination of basic statistical discontinuity parameters, such as normal set spacing, linear frequency, trace length, and termination from linear scanline surveys, by applying the open-source software Octave is presented. The theoretical background for this workflow has been established in the last few decades by various authors and is summarized in [14]. With the help of an example data set, the workflow is presented step by step to the interested reader to further push the application of the scanline method in the engineering geology community. The large amount of time required for in situ measurements and subsequent analyses is probably the reason that scanline measurements are still not carried out as a standard. The aim of this contribution is to keep the entry threshold as low as possible and to minimize the time required for the evaluation of scanline data considerably. Therefore, all Octave scripts of the workflow are provided for the processing of the workflow and application to other case studies.

**Supplementary Materials:** The following supporting information can be downloaded from https://www.mdpi.com/article/10.3390/app12199622/s1. For each of the six steps data-, m- and mat-files are provided, which are stored in the following folders: Step1_scanline_data, Step2_cos, Step3_filtering, Step4_spacing, Step5_trace_length_termination, Step6_spacing_histogram and Step6_trace_length_histogram.

**Author Contributions:** Conceptualization, C.Z. and T.S.; methodology, C.Z.; software, C.Z., T.S., and M.K.; validation, T.S. and M.K.; formal analysis, T.S.; investigation, M.K.; data curation, C.Z.; writing—original draft preparation, C.Z.; writing—review and editing, T.S. and M.K.; supervision, C.Z.; project administration, C.Z.; funding acquisition, C.Z. All authors have read and agreed to the published version of the manuscript.

**Funding:** This study was part of the alpS research project AdaptInfra, which was supported by TIWAG, ILF Consulting Engineers, and the Austrian Research Promotion Agency (COMET-program). The alpS-K1-Centre was supported by the federal ministries BMVIT and BMWFW, as well as the states of Tyrol and Vorarlberg in the framework of COMET—Competence Centers for Excellent Technologies. COMET is processed through FFG.

**Data Availability Statement:** All data and m-files used for this study are available at Supplementary Materials. This program is free software: you can redistribute it and/or modify it under the terms of the GNU General Public License as published by the Free Software Foundation, either version 3 of the license or (at your option) any later version. This program is distributed in the hope that it will be useful, but without any warranty; without even the implied warranty of merchantability or fitness for a particular purpose. See the GNU General Public License for more details (http://www.gnu.org/licenses/ accessed on 1 March 2022).

**Acknowledgments:** The authors wish to acknowledge alpS for supporting this study in the framework of the research project AdaptInfra. The quality of the manuscript was improved by the constructive comments of three anonymous reviewers.

**Conflicts of Interest:** The authors declare no conflict of interest.

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
