# Peer review of "Semiautomated Statistical Discontinuity Analyses from Scanline Data of Fractured Rock Masses"

_applsci, doi:10.3390/app12199622_

Round 1
Reviewer 1 Report
This manuscript discussed about semi-automated statistical discontinuity analyses from a scanline data by applying an open-source software, Octave. The idea discussed in this manuscript is actually simple, but it is helpful to handle the processing of scanline data. The authors should discuss the similar methods and also the advantage and disadvantage of methods. The borehole discontinuity imaging, if any, can be added for comparison beside scanline surveys.

Author Response
Dear Reviewer,
our comments are attached as pdf.
Best regards
Christian Zangerl

Reviewer 2 Report
R. Cox’s review of ‘Semi-automated statistical discontinuity analyses from scan-2 line data’ by Zangerl et al. for Applied Sciences, August 2022
The Zangerl et al. manuscript presents a step-by-step software-aided procedure to compute a 3-D fracture network in a non-stratified rock body. The manuscript is well written, organized and referenced and will be a good contribution to the field.
I recommend publication with minor revision.
Here is a list of minor suggestions (in red) for clarity and readability:
Lines 2 -3, ‘Semi-automated statistical discontinuity analyses from scan-line data of non-stratified rock bodies’
Line 24, ‘materials in order to minimize initial training time’
Line 57, ‘character, outcrop’
Line 58, change ‘accomplished’ to ‘complimented’
Line 101, ‘time and to further’
Line 176, ‘discontinuity sets not only on the basis’
Line 181, ‘mass (e.g. [2, 28, 30]). A too’
Lines 229-230, ‘probability functions, for example the log-normal distribution or the Weibull distribution, may be’
Line 254, define ‘TLS’
Line 278, Should this say ‘Equation 9’?
Line 300, ‘implemented a 6th-order polynomial function in the order of 6 to fit a’
Line 345, ‘the difference between each value in the column is calculated’
Line 430, ‘with a negative exponential’
Lines 447-448, ‘class interval should be defined as proposed by approach III, respectively.’
Line 486, ‘rock quality designation (RQD)’
Line 517, ‘study area can, if at all, only be rudimentary captured’
Lines 521-522, ‘sample size, the reader is referred to [14, 34], and for discontinuity size estimations to [14, 36], respectively.’
Figure 1, Perhaps include columns for ‘mineralization’ and ‘opening width’. For stratified rocks, columns for ‘strike and dip of strata’ and ‘strata thickness’ would be needed.
Figure 4, ‘d)’ is mislabeled ‘c)’
Author Response

(The authors gave the same response as above.)

Reviewer 3 Report
This paper presents a workflow for the semi-automatic determination of basic discontinuity parameters by applying the open-source software Octave. The paper needs major revision and some comments should be addressed.
1. The paper focuses on the work process, and should highlight the research innovation points. Please reorganize the structure to highlight the research points of the paper.
2. The method in this paper should be compared with other methods or software.
3. The research content and results are highlighted in the abstract, the description of supplementary materials can be omitted.
Author Response

(The authors gave the same response as above.)

Round 2
Reviewer 1 Report
Thank you for revising the manuscripts and reply to my comments. I finally agree that this manuscript is accepted for publication in Applied Sciences.
Reviewer 2 Report
This manuscript is improved adequately.